# Comparative Characterization of NWFET and FinFET Transistor Structures Using TCAD Modeling

**DOI:** 10.3390/mi13081293

**Published:** 2022-08-11

**Authors:** Konstantin O. Petrosyants, Denis S. Silkin, Dmitriy A. Popov

**Affiliations:** 1School of Electronic Engineering, HSE University, 34 Tallinskaya St., 123458 Moscow, Russia; 2Institute for Design Problems in Microelectronics of Russian Academy of Sciences, 9 Prospekt 60-Letiya Oktyabrya, 117312 Moscow, Russia; 3School of Computer Engineering, HSE University, 34 Tallinskaya St., 123458 Moscow, Russia

**Keywords:** fin-shaped transistor FinFET, nanowire NWFET, TCAD-simulation, electro-thermal effects

## Abstract

A complete comparison for 14 nm FinFET and NWFET with stacked nanowires was carried out. The electrical and thermal performances in two device structures were analyzed based on TCAD simulation results. The electro-thermal TCAD models were calibrated to data measured on 30–7 nm FinFETs and NWFETs. The full set of output electrical device parameters I*_on_*, I*_off_*, SS, V*_th_*, and maximal device temperature Tmax was discussed to achieve the optimum VLSI characteristics.

## 1. Introduction

In the era of information technology, high speed, low power and multi-functional devices supporting high density Integrated Circuits (ICs) are essential for micromachine systems. The very large scale ICs (VLSIs) have advanced over recent years with increasing the number of on-chip transistors every generation and reducing the device sizes. Metal-Oxide-Semiconductor Field Effect Transistors (MOSFETs) are the fundamental components of the Very Large ICs. With the development of Moore’s law the transistor on the VLSIs chip count is doubled every 2 years (see Figure 1) [1]. Because of the exponential increase in gate leakage current, the scaling of planar MOSFETs has encountered a fundamental limitation below the 65 nm technology node. To overcome this obstacle, the silicon-on-insulator (SOI), ultrathin body (UTB), and high-k dielectric have been introduced in 45–32 nm node production [2]. To bring 22–7 nm node devices into industrial production, the Si non-planar transistor structures were introduced as a possible replacement for the planar MOSFETs. Among these devices, the fin-shaped FETs (FinFETs) and nanowire FETs (NWFETs) have been enabled.

Successfully continuing the Moore’s law using downscaling of CMOS technology to the sub-10 nm nodes [3,4,5], a retrospective review of logic MOSFET structures scaling is presented in Figure 2. It is seen that every subsequent transistor’s structure improves the electrostatic control of the device channel and leads to a higher operation current I*_on_*/I*_off_* ratio [6].

In recent years "three-dimensional transistors" FinFETs have firmly established themselves as basic elements for the fabrication of modern VLSIs using 28–7 nm technologies. The FinFETs are basic components of industrial microprocessor series [7,8,9,10,11,12].

Cross-sectional TEM pictures of FinFET structures are presented in Figure 3. The technology process and device characteristics’ descriptions were published in [13,14,15,16,17], and TCAD modeling and simulation results were published in [18,19,20,21].

At the same time, the progress in the VLSI CMOS technology has allowed to fabricate in the sub-14 nm nodes a new viable solution—nanowire FET (NWFET) with vertically-stacked NWs, which realizes a "three-dimensional packaging of transistors” (see Figure 3c) [5]. NWFETs provide better layout area efficiency than FinFETs. Gate-all-around (GAA) nanowire (NW) FETs with the gate fully wrapped around the device body offer superior gate electrostatic control.

We focused on the properties of NWFET which are different from those of 3D FinFET.

The gate-all-around configuration of NWFET is the ideal architecture for channel control. In comparison with FinFET, it relaxes channel film thickness requirements for a target leakage control. The stacking channel yields high current levels per layout surface overcoming the current limit imposed by small width/pitch ratio in FinFET structures. In [24] the co-processed NWFET (W*_Si_* = 15 nm) and FinFET (W*_Si_* = 20 nm) fabricated on SOI wafers were compared. It was shown experimentally that, in NWFET structures, the DIBL parameter (Drain Induced Barrier Lowing–main factor stimulating the small-size effects in MOSFETs) was strongly reduced in comparison with FinFET (see Figure 4). It is, however, comparable to 6-nm FinFET. Moreover, measured I*_d_*-V*_d_* and I*_d_*-V*_d_* characteristics demonstrate the value of on-state current I*_on_*: 6.5 mA/um (n-MOS) and 3.3 mA/um (p-MOS) for V*_g_* = 1.2 V that are 2.5 times higher than for FinFET. However, the carrier mobility in the NWFET is smaller than in the FinFET (see Figure 5). It indicates that carrier transport in nanowire is limited by lateral factors, e.g., series resistance, potential barriers at source/drain, etc. Moreover, Figure 5 shows that FinFET’s effective mobility is strongly decreased with Si width W*_Si_* reduction. This effect is attributed to a degraded Si-SiO_2_/HfO_2_ interface when W*_Si_* is reduced down 7–5 nm.

NWFETs comprise the vertically-stacked nanowires to achieve better layout area efficiency than a FinFET. At sub-14 nm nodes, outflow of the heat generated at the channel-drain junction is impeded due to low thermal conductivity materials in the device structure, e.g., SiO_2_ and thin layers of silicon. Consequently, self-heating effects (SHE) change the normal values of carrier mobility, saturation velocity, and interface trap density, resulting in device parameters’ degradation [25]: threshold voltage V*_th_*, subthreshold swing SS, drain current I*_on_*, I*_off_*. It was shown by TCAD simulation that the maximal channel temperature increase in stacked 14 nm NWFET can rise to about 70–80 °C for the device-on-bulk substrate and to about 160 °C above ambient for the device on SOI substrate when 200 µW power was applied [24]. High local temperature leads to device reliability and operating lifetime reduction [26].

In spite of NWFETs representing the next step of scaling, they were not used immediately in the industrial serial fabrication process. Further technology and device optimization was necessary to guarantee desired integrity, performance, and reliability along with compatibility to conventional CMOS technology. The leading microelectronics fabs are planning to start with the NWFET based VLSIs industrial fabrication in the nearest future [27,28,29].

Nevertheless, even now the gate-stack silicon NWFETs have an increasing demand in medical, agricultural, and military systems where micromachines are used. The nanowire FETs have applications particularly in logic integrated circuits, SRAM memory, sensor and measurements electronics.

For the logic ICs based on NAND/NOR or XOR/XNOR logic operators NWFET technology has provided, on average, 13.9% smaller and 29.5% faster circuits then in FinFET CMOS technology at 22 nm technology node [30].

The SRAM 6T-bitcells build on NWFETs demonstrate in comparison with FinFETs the lower parasitic RC-delay, reduced power consumption, 30% decrease of cell area, smaller minimum operating voltage (V*_min_* = 0.6 V) and lower standby leakage current value at 5 nm node design rules [31,32].

To realize fully integrated nanoprocessor with computing, memory and addressing capabilities two interconnected programmable NWFET transistor arrays were developed with 496 functional FET nodes in the area of 960 µm^2^. NWFET devices with 10-nm diameter SiGe channel and stacked-gate geometry were used. By mapping different active node patterns into the array, logic functions including full adder, full-subtractor, multiplexer, demultiplexer and D-latch can be realized [33].

The typical example of using NWFETs in sensor technique is the detection of biomolecular species [34,35,36]. The medical sensors based on high-performance NWFET’s arrays fabricated on transparent quartz substrates facilitate high-resolution optical imaging at both substrates of the samples under test and were used for mapping neural circuits in acute brain slices [37]. The active area of typical NWFETs, 0.06 µm^2^, was much more locally sensitive and, thus provided better spatial resolution of signals compared to planar FETs that were 60 µm^2^.

Therefore, it is necessary to compare from the electro-thermal standpoint the FinFET and NWFET transistor structures with the same footprint to achieve the optimum VLSI architecture fabricated by sub-14 nm technologies.

It is problematic to evaluate the impact of device dimensional scaling using only experimental studies. A systematic simulation study based on deep semiconductor device physics is still lacking to explain the observed trends. For this purpose using the TCAD modeling is the most effective way. Simulation of FinFETs have been reported in [38,39,40]. Most papers benchmarking stacked NWFETs have been carried out using drift-diffusion based models [41,42,43] and did not take into account some physical effects important for nanometer 3D structures. All the works mentioned above were focused on specific physical effects in FinFET and NWFET structures and did not contain the complete set of most important output electrical device parameters I*_on_*, I*_off_*, SS, V*_th_*.

In this paper the complete comparative characterization of FinFET and NWFET is carried out. The electrical and thermal performances in both device structures are analyzed based on TCAD simulation results. For this purpose the electro-thermal TCAD models were calibrated to data measured on sub-14 nm FinFETs and NWFETs. The full set of the output device parameters I*_on_*, I*_off_*, SS, V*_th_*, T*_max_* is analyzed.

## 2. Device under Test. TCAD Setup

The stacked nanowire NWFET structure on SOI substrate similar to Figure 3c with parameters given in [44] was selected for consideration. The parameters of device structure are as follows: the length of the transistor channel is 100 nm, the width is 14 nm, the gate dielectric is HfO_2_ with an equivalent oxide thickness of 1.8 nm, the gate metal is TiN. The doping level of the drain and source areas is assumed to be 2 × 10^20^ cm^−3^, the channel area is −5 × 10^18^ cm^−3^.

For the comparison of electro-thermal performance the equivalent FinFET structure with the same footprint must be taken. In [45] the FinFET with size ratio W:H = 1:2 was compared with two stacked NWs with H = W. The FinFET fin height H was chosen equal to the sum of two NWFET nanowire widths W. We consider that this criterion of equivalence is not quite correct: in the NWFET structure all four sides of the channel—wire perimeter are governed by the gate. The fact that the bottom side of the FinFET channel is not governed by the gate was not taken into account.

The active perimeter of three nanowires with a side of 14 nm is 168 nm = 14 nm × 4 × 3. The active perimeter of a FinFET with a width of 14 nm and a fin height of 42 nm is 98 nm = 42 nm × 2 + 14 nm. In order to obtain the same current value in the FinFET model, the fin height of a FinFET must be chosen as 77 nm to provide an active perimeter of 168 nm = 77 nm × 2 + 14 nm. Therefore, we have taken for comparison with NWFET in Figure 6a the equivalent FinFET in Figure 6b with increased size ratio 1:5.5 (instead of 1:3) to provide the drain current equivalence for the device structures under test.

To study the electrical and thermal performances of stacked NWFETs and FinFETs, 3D self-consistent electro-thermal device simulation was performed using TCAD Sentaurus Synopsys simulator [46]. The following models of physical effects were used:Charge Carrier Transport and Temperature-Hydrodynamic Model.Quantization-Density Gradient Model.Recombination Model-SRH (DopingDep).BandGap Narrowing Model-Slotboom.

Mobility description includes several degradation models. Degradation due to scattering of charge carriers on impurity atoms and other charge carriers is described by the Philips Unified Mobility Model [47] (PhuMob). The mobility model of the inversion and storage layers (IALMob) determines the degradation of mobility at the dielectric-semiconductor interface [48]. This model includes Coulomb scattering of impurities, phonon scattering, and surface roughness scattering. The Ballistic Mobility model is included to account the effects of a short channel [49]. The contribution to the mobility of different models is combined according to the Matthiessen’s rule: (1)$$\frac{1}{\mu }=\frac{1}{\mu_{IALMob}}+\frac{1}{\mu_{PhuMob}}+\frac{1}{\mu_{BalMob}}$$

Similar sets of physical models were used in publications devoted to TCAD modeling of sub-14 nm FinFETs and stacked NWFETs [24,43,50]

In order to validate simulation accuracy, the TCAD device models were calibrated to the data measured from sub-100 nm bulk [51] and 15 nm SOI [52] n-channel FinFETs.

This set of models was previously used to model FinFET structures with a channel length from 15 nm to 60 nm and a fin width from 5 nm to 35 nm. The TCAD models developed on its basis had good accuracy, and additional calibration was used only to adjust the threshold voltage and external resistance, if such a need arose.

## 3. Electrical Simulation Results

The geometrical model of the stacked nanowire NWFET and equivalent FinFET structures are shown in Figure 6. The levels of doping, the thickness of the gate dielectric, the work of the electron output in the metal, as well as the models of physical effects used in device structures were identical. A comparison of the simulated drain-gate I*_d_*-V*_d_* characteristics with the measured results given in [44] is shown in Figure 7. The error does not exceed 15%.

The comparison for two types of FETs with the same structure parameters was carried out according to the following parameters: leakage current I*_off_* at the gate voltage of 0 V, maximal on-state current I*_on_* at the gate voltage of 2 V, the sub-threshold swing SS measured on the linear section of the characteristic, as well as the threshold voltage V*_th_* at the current value of 0.1 µA. In all cases, the drain voltage was 1.2 V.

The comparison results are summarized in Table 1. NWFET demonstrates a higher value of drain currents I*_on_* and I*_off_*, both maximal on-state and leakage currents, as well as a lower value of threshold voltage V*_th_* and sub-threshold swing SS. This is primarily due to the fact that in NWFET the gate completely surrounds the channel semiconductor. As a result, a channel current flow is formed along the entire perimeter of the semiconductor. At the same time, due to the small size of the nanowire, the channel area extends over almost its entire cross-sectional area, and the current inside is distributed more evenly than in a FinFET of similar sizes.

A comparison of the current density distribution in FinFET and NWFET structures in Figure 8 shows that the current density in FinFET is extremely unevenly distributed. The current density in the central part of the fin is two to three orders of magnitude lower (yellow area in the figure) than the current density at the walls. In NWFET, more than half of the area of the “nanowire" passes through significant current densities. More efficient use of the available area leads to the fact that the NWFET requires less voltage at the gate for switching to the open state, while it conducts current ions. However, the same effects lead to an increase in leakage current I*_off_*. The value I*_off_* = 146 pA (see Table 1) is too much. This is the poor feature of NWFET. To reduce the leakage current, we must “shift" the I*_d_*-V*_gs_* characteristics in Figure 7 to the right.

There are several ways to minimize the I*_off_* value. In the ordinary common gate NWFET structure Figure 3c this can be achieved by increasing the dopant concentration N*_ch_* in the channel region from $5\times 10^{18}$ cm^−3^ to $8\times 10^{18}$ cm^−3^. The value of leakage current I*_off_* falls down up to 8 pA (see Table 2).

The small I*_off_* values were achieved in the independent gate ΦFET structures. The electrostatic control I*_off_* values in ΦFETs were achieved 1–2 decades lower than in individual gate FinFET [44].

In real cases from industry the FinFET should not have a vertical shape of the sidewall, and the NWFET nanowire channel geometry is not rectangular. This fact also affects the transistor parameters, as shown in our work [52]. The FinFET sidewall angle of 18 degrees increased SS by 8.3% and V*_th_* by 4.2%, also lowered I*_on_* by 12.9%.

For the NWFET the influence of the NW corner rounding effect is illustrated in Figure 9. It was shown using TCAD simulation that the corner rounding effect with radius R = 2.5 nm caused the I*_on_* current decrease: by 10% for W*_nw_* = 14 nm; by 19% for W*_nw_* = 8 nm and by 33% for W*_nw_* = 5 nm in comparison with appropriate results received for square channel (V*_ds_* = 1.2 V, V*_gs_* = 2 V for all cases). Our estimations agree well with the results reported in [20].

A special rounding coefficient (CR) was introduced to take into account the corner rounding effect in NWFETs [53]. The I*_d_*-V*_d_* curves for the device with L = 20 nm and D*_Si_* = 8 nm are shown in Figure 10. It was shown that the on-state current decreases by 20% due to rounding effect. Figure 10 also shows that oxide thickness engineering is more efficient than the shape of nanowire cross-section profile [see Table 3].

## 4. Thermal Simulation Results

It is necessary to take into account that, in nanosize FinFETs and NWFETs, the high drain current density causes a high power density, which contributes to overheating of the device. In this regard, studies of the thermal properties in NWFETs and FinFETs and their reliability are of particular relevance in this area.

Stacked NWFET are expected to be in the mainstream for sub-10 nodes, so they could be expected to suffer even more severely from SHE due too smaller cross-section on NWs compared to fins [24].

The thermodynamic model was used as a thermal model. The ambient temperature was chosen as 300 K. The used thermal TCAD models were verified in practical simulation of FinFET structures with different layouts [54]. The results were compared with the results presented in [24,45] and shown an acceptable agreement.

Images of the heat distribution over the FinFET and NWFET structures are shown in Figure 11. It follows from the figure that NWFET nanowires located in the depth of the structure are slightly worse at removing heat than those located near the surface. However, due to the fact that each nanowire is surrounded by polysilicon, which has a higher thermal conductivity than the materials of the gate stack, the difference between different floors is minimal.

Due to the small area and more uniform current distribution, the nanowire placed inside of device structure is heated almost evenly. At the same time the heat flow from the upper faces of the nanowires is more intensive due to the shorter distance to the thermal contact. However, the effect is hardly noticeable: the temperature difference at the lower and upper faces is only 1 K.

The maximum temperature in FinFET is observed at points with the maximum current density. In FinFET, the temperature difference inside fin area can reach up to 10–15%.

An important role in determining the thermal characteristics plays the external thermal resistance, which depends on the structure of BEOL, as well as the features of the casing. In [24], the thermal resistance was measured as 1.46 × 10^6^ K/W for 14 nm SOI FinFET, similar in size and structure to the devices considered in this paper. To assess the effect of BEOL and the casing on the device self-heating, a simulation of the I*_d_*-V*_gs_* characteristics taking into account this thermal resistance, was carried out.

The main electrical parameters and maximum temperature T*_max_* for FinFET and NWFET due to overheating are summarized in Table 4.

It is seen in Figure 12 that the effect of self-heating becomes noticeable when the drain current exceeds a value of about 30 µA. It means that only the parameter I*_on_* is changed due to SHE. Other electrical parameters are defined at the sub-threshold mode and do not depend on SHE.

Adding an external thermal resistance to the model significantly increases the internal temperature of the device structure. The temperature distributions taking into account the thermal resistance are shown in Figure 11. It is seen that the presence of external thermal resistance leads to an increase of maximum temperature inside the FinFET and NWFET structures about to 60 K in comparison with the simulation results when the thermal resistance was not taken into account.

For practical applications it is interesting to compare the temperature regimes in the multiple-gate cells based on FinFETs and vertically-stacked NWFETs in the case of maximal power dissipation. The results are presented in Figure 13. In FinFET structure all three fins:(1)dissipated the heat toward the thermal contact;(2)operate in the identical thermal condition with similar temperature.

In the case of NWFET:(1)only the upper wire can dissipate straight toward the thermal contact;(2)the lower wires situated in the depths of construction are heated more than the upper wire.

It is seen that for the same power dissipation about 0.24 mW the maximal temperatures are: 366 K for the FinFET cell and 402 K for the NWFET cell. These results confirm the fact that the NWFET cells with vertically-stacked wires construction have the problem with heat outflow in high power mode.

It follows from Figure 13 that three transistors with similar structures and with the same values of external thermal resistance and conducted power, placed on common horizontal substrate platform and fabricated using FinFET technology, experience significantly less overheating in comparison with NWFET technology.

It is interesting to compare our results with the results presented in [24]. For 14-nm SOI FinFET with applied power 0.225 W the temperature rises by about 160 K above ambient looks too high in comparison with 66 K in our work.

## 5. Conclusions

A complete comparison for 14 nm FinFET and NWFET with stacked nanowires was carried out. The electrical and thermal performances in two device structures was analyzed based on published works review and our own TCAD simulation results. The electro-thermal TCAD models were calibrated to data measured on 30–7 nm FinFETs and NWFETs. The full set of output electrical device parameters I*_on_*, I*_off_*, SS, V*_th_*, and maximal device temperature T*_max_* was discussed to achieve the optimum VLSI characteristics.

It was shown that in the NWFET structure the drain current density distribution in the nanowire channel cross-section is more compact than in the fin cross-section of the equivalent FinFET-structure. It means the larger value of drain open-mode current Ion in NWFET in comparison with FinFET. It is advantage but at the same time it means larger value of leakage current I*_off_*. A special efforts must be taken to press the high level of I*_off_*.

Electro-thermal modeling of FinFET and NWFET structures was carried out. It was shown that the maximal temperature in 14 nm-NWFET structure is 35 K more than in equivalent FinFET structure. It means that the heat dissipation in sub-14 nm NWFET structures is a problem that requires a constructive solution.

## Figures and Tables

**Figure 1 micromachines-13-01293-f001:**
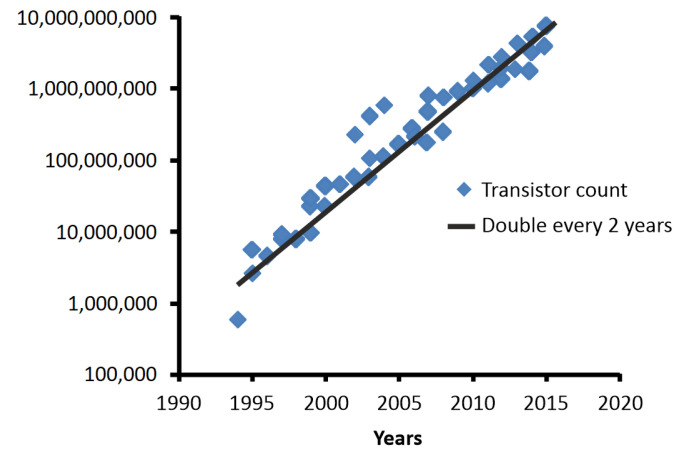
Chronological interpretation for last 30 years of the Moore’s law: transistor count on the VLSI chip is doubled every 2 years because the sizes of transistors are decreased.

**Figure 2 micromachines-13-01293-f002:**
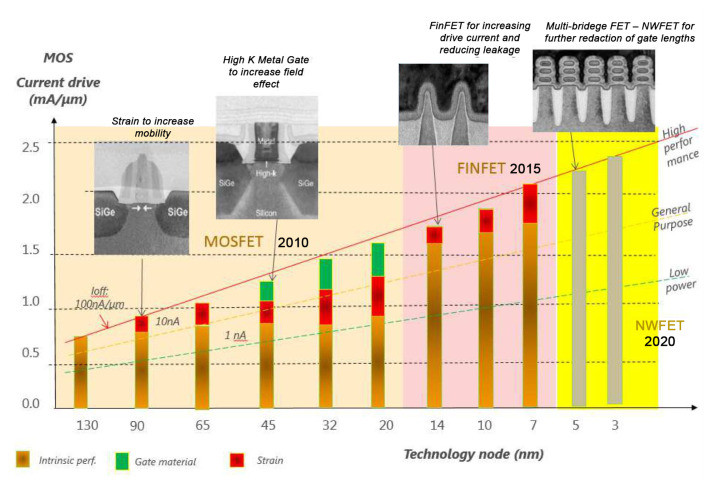
Retrospective review of logic MOSFET structures scaling. The ratio of drive current I*_on_* to leakage current I*_off_* is steady increased for each more advanced transistor generation.

**Figure 3 micromachines-13-01293-f003:**
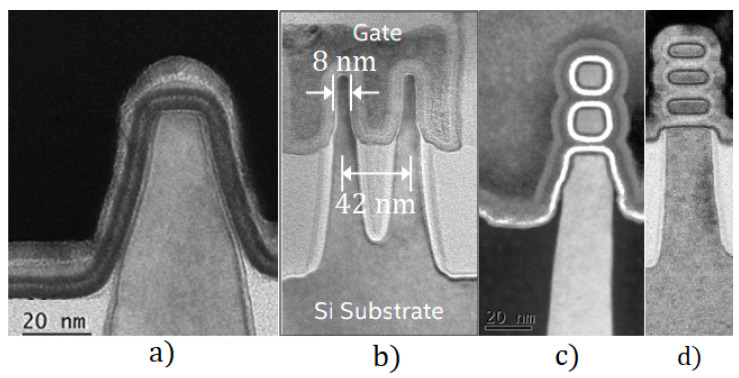
Cross-sectional TEM pictures of novel 3D structures: (**a**) ordinary bulk sub-100 FinFET structure [13] (**b**) 14-nm FinFET by Intel [15] (**c**) 15-nm Ge NWFET [22] (**d**) 5-nm NWFET by IBM [23].

**Figure 4 micromachines-13-01293-f004:**
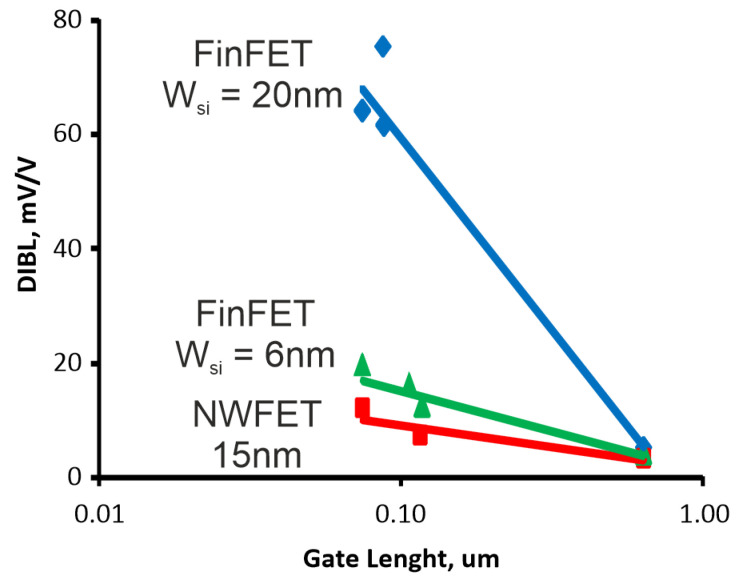
DIBL versus gate length L. NWFET has a lower DIBL than FinFET.

**Figure 5 micromachines-13-01293-f005:**
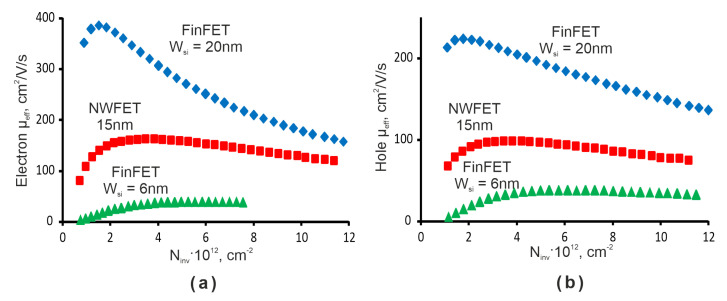
Electron (**a**) and hole (**b**) effective mobility vs. the inversion charge density (L = 600 nm).

**Figure 6 micromachines-13-01293-f006:**
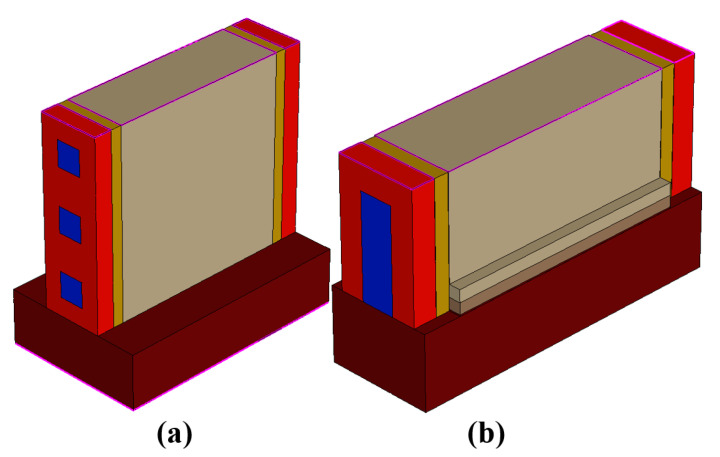
(**a**) NWFET structure in the TCAD model; (**b**) FinFET structure in the TCAD model.

**Figure 7 micromachines-13-01293-f007:**
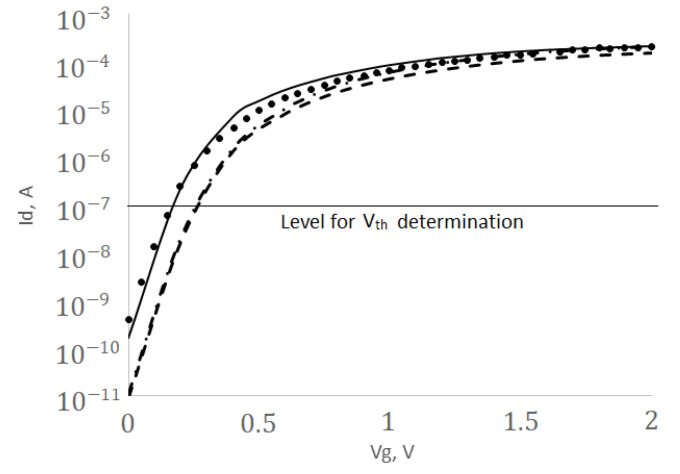
Drain-gate I*_d_*-V*_gs_* characteristics: the dots indicate experimental data of NWFET, the main line is the results of NWFET TCAD modeling, the dotted line is the results of FinFET TCAD modeling (fin height 42 nm), the dashed line is the results of FinFET TCAD modeling with an increased fin height 77 nm; for all cases V*_ds_* = 1.2 V.

**Figure 8 micromachines-13-01293-f008:**
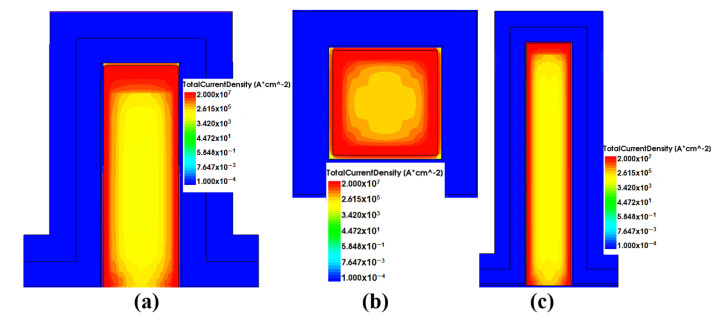
Simulation results; (**a**) current density distribution in the FinFET (height 42 nm) channel; (**b**) current density distribution in the NWFET channel; (**c**) current density distribution in the FinFET (height 77 nm).

**Figure 9 micromachines-13-01293-f009:**
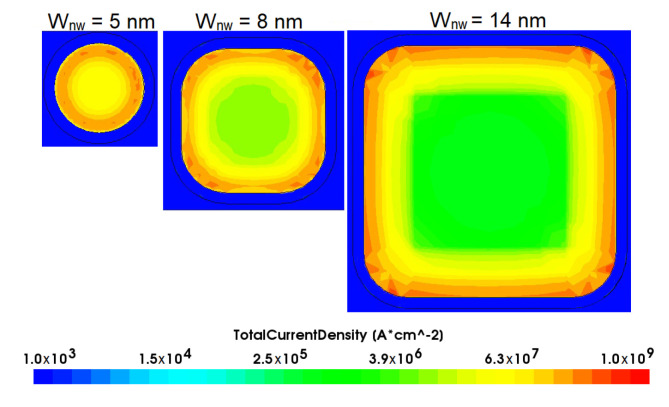
Simulated current density profile at a cross-sectional plane perpendicular to the channel of n-NWFET for varying W*_nw_* size.

**Figure 10 micromachines-13-01293-f010:**
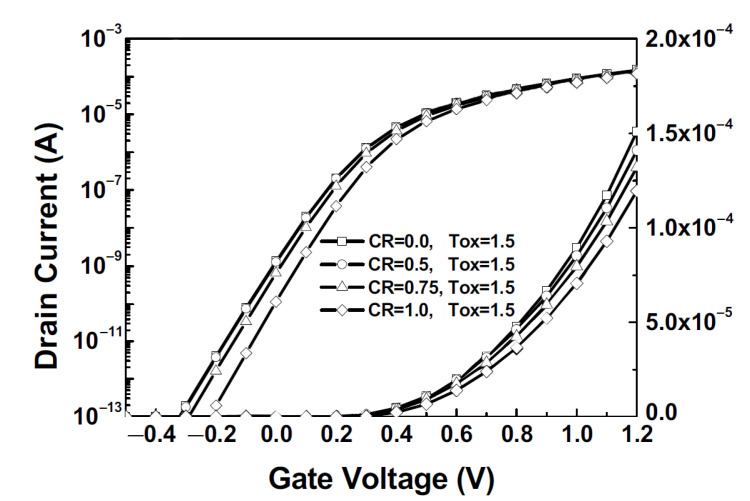
I*_d_*-V*_g_* characteristics with respect to corner rounding and T*_ox_*; CR is corner rounding ratio 2R/W*_nw_*, where R rounding radius.

**Figure 11 micromachines-13-01293-f011:**
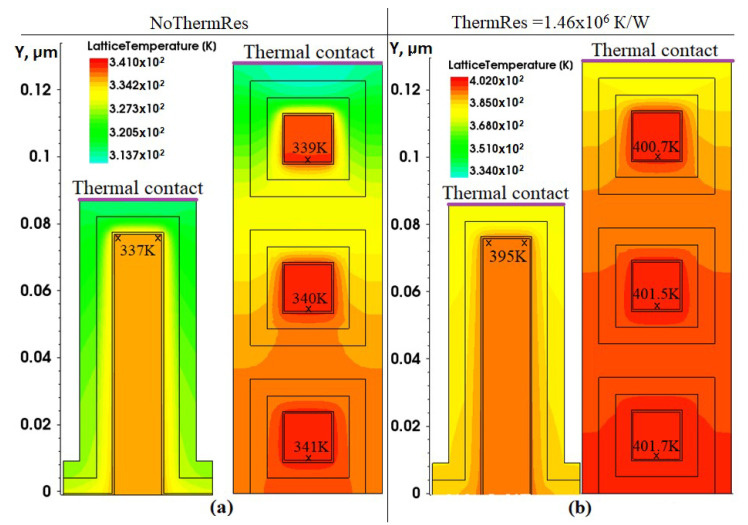
Temperature distributions: (**a**) in the FinFET structure; (**b**) in the NWFET structure.

**Figure 12 micromachines-13-01293-f012:**
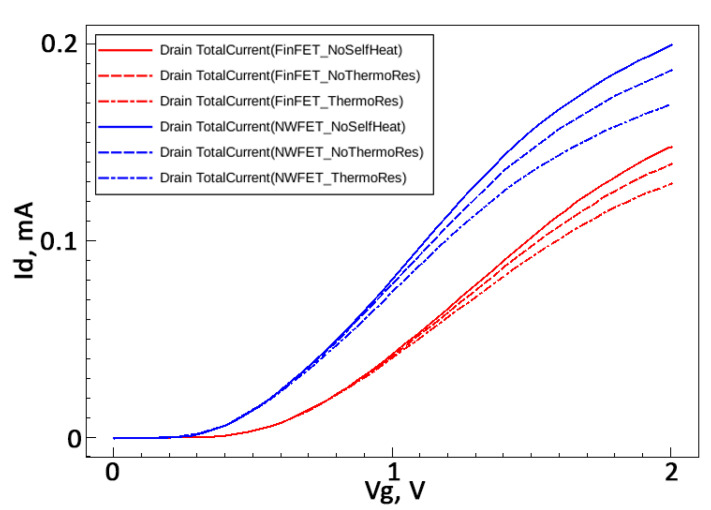
Comparison of FinFET and NWFET I*_d_*-V*_g_* characteristics taking into account self-heating.

**Figure 13 micromachines-13-01293-f013:**
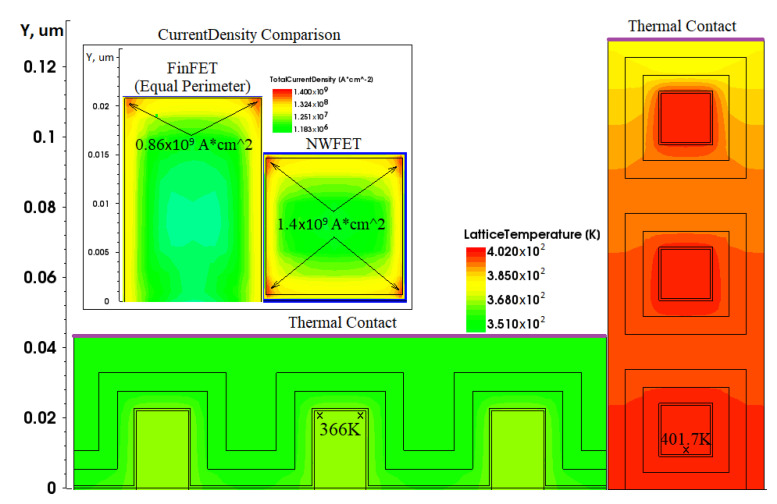
Comparison of the temperature distributions in 3-fin FinFET and NWFET cells.

**Table 1 micromachines-13-01293-t001:** Comparison of the main parameters of NWFET and FinFET.

Type	I*_off_*, pA	SS, mV/Decade	V*_th_*, mV	I*_on_*, µA
NWFET (each side 14 nm)	146	64	178	200
FinFET (fin height 42 nm)	9	70	258	148
FinFET (fin height 77 nm)	12	70	252	195

**Table 2 micromachines-13-01293-t002:** Comparison of options to reduce the leakage current by changing various parameters of the NWFET structure.

NWFET Structure Parameters	I*_off_*, pA	SS, mV/dec	V*_th_*, mV	I*_on_*, µA
N*_ch_* = $5\times 10^{18}$ cm^−3^, EOT = 1.8 nm, WF = 4.205 eV, W*_nw_* = 14 nm	146	64	178	200
N*_ch_* = $8\times 10^{18}$ cm^−3^, EOT = 1.8 nm, WF = 4.205 eV, W*_nw_* = 14 nm	8	62	248	177

**Table 3 micromachines-13-01293-t003:** Subthreshold swings as a function of Tox (SS, Subthreshold swing mV/dec).

T*_ox_*	1.5 nm	2.0 nm	2.5 nm	3.0 nm	3.5 nm
SS	76.7	85.5	96.7	111.2	130.1

**Table 4 micromachines-13-01293-t004:** The effect of overheating on the main parameters of transistors.

Type	Thermal Conditions	Tmax, K	I*_off_*, pA	SS, mV/Decade	V*_th_*, mV	I*_on_*, µA
	No SelfHeating	300	146	64	178	200
NWFET	No ThermRes	341	146	64	178	187
	ThermRes 1.46 × 10^6^	402	146	64	178	170
	No SelfHeating	300	9	70	258	148
FinFET	No ThermRes	328	9	70	258	139
	ThermRes 1.46 × 10^6^	371	9	70	258	130

## Data Availability

The data presented in this study are available in articles listed in "References" section.

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
