# Peer review of "Comparative Characterization of NWFET and FinFET Transistor Structures Using TCAD Modeling"

_micromachines, 2022, doi:10.3390/mi13081293_

Round 1
Reviewer 1 Report
Q1. In line 71, author mentioned the fin height had to be 71 increased to 77 nm, which is 1.83 times more than the original one to obtain the same current value. I have no idea why the Figure.2 shows a current difference at the linear region? Can you please explain about it?
Q2. In table.1, author summarized the I-V parameters from TCAD simulations. However, a Vth shifting can be observed from the summary. I suggest to add some descriptions or discussions about the Vth shifting vs NW size correlation.
Q3. In Ioff discussion, the high Ioff was attributed to the total area effect. However, the Ion will be reduced and then smaller than FinFET case. I'm confused why the doping concentration should be changed?
Q4. In thermal simulations, did author consider about the bulk Si substrate thermal conductivity?
Q5. Suggest to modify the resolution of Fig.5, the number is too small that hard to read in the manuscript.
Q6. In real case from industry, the FinFET should not be a vertical shape of the sidewall. What will be different to the vertical sidewall versus tapper sidewall?

Author Response
Dear Reviewer
We thank You very much for Your remarks and suggestions. All Your remarks and suggestions were taken into account:
- the Introduction part with literature review was prepared
- the number of pages was increased from 7 to 13
- the number of figures: from 6 to 14
- number of references: from 14 to 47
The Reviewer’s comments |
Author’s responses |
Q1. In line 71, author mentioned the fin height had to be 71 increased to 77 nm, which is 1.83 times more than the original one to obtain the same current value. I have no idea why the Figure.2 shows a current difference at the linear region? Can you please explain about it? |
The equivalent H=77 nm in FinFET was introduced to achieve the same value only for Ion in saturation mode. The current difference in the linear region and sub-threshold is observed due to subthreshold voltage shift |
Q2. In table.1, author summarized the I-V parameters from TCAD simulations. However, a Vth shifting can be observed from the summary. I suggest to add some descriptions or discussions about the Vth shifting vs NW size correlation. |
It was partially done in the introduction and in Fig. 5, Fig. 10 and Fig. 11 |
Q3. In Ioff discussion, the high Ioff was attributed to the total area effect. However, the Ion will be reduced and then smaller than FinFET case. I'm confused why the doping concentration should be changed? |
Channel doping increasing is one of the ways to decrease Ioff in 1-2 orders of value. The Ion decrease is not so considerable – about 10%. The best Ion/Ioff ratio was achieved in ФFET structure (Fig. 3b) |
Q4. In thermal simulations, did author consider about the bulk Si substrate thermal conductivity? |
It is well known that SOI FinFET and NWFET suffer more from self-heating than bulk ones. So, this type of devices was chosen for consideration. The Reviewer’s remark is true: thermal conductivity of Si-substrate is 100 times higher than SiO2 substrate conductivity. |
Q5. Suggest to modify the resolution of Fig.5, the number is too small that hard to read in the manuscript. |
Fig. 5 was improved. |
Q6. In real case from industry, the FinFET should not be a vertical shape of the sidewall. What will be different to the vertical sidewall versus tapper sidewall?
|
The discussion of this problem was added in the manuscript. |
With best regards, sincere Yours
Konstantin O. Petrosyants.

Reviewer 2 Report
In this manuscript, “Comparative Characterization of NWFET and FinFET Transistor Structures using TCAD Modeling”, the authors have performed a comparison for 14 nm FinFET and NWFET with stacked nanowires. For this purpose the electrical and thermal performances in two device structures were analyzed based on TCAD simulation results. Overall the work is novel and can be accepted for publication after these minor recommendations.
1) Authors should explain further information about stacked nanowire-based field effect transistor. How it works and some review of literature.
2) In line 17, the authors state that stacked NWFET construction causes the degradation of device parameters: carrier mobility μe f f , threshold voltage Vth, drain currents, Ion/Io f f. It will be better if the authors add one paragraph regarding the theoretical discussion that how these device parameters are degraded for stacked NWFT construction.
3) Authors should explain the models used in this simulation such as Charge Carrier Transport and Temperature - Hydrodynamic Model. Quantization - Density Gradient Model. Mobility Model - Enormal (Ialmob Coulomb2D) PhuMob. Ballistic Mobility - BalMob. Recombination Model - SRH (DopingDep). BandGap Narrowing Model - Slotboom.
4) It will be better to draw a flow chart for the simulation.
5) Table 1: why threshold (as well Ioff) varies so much from one device to the other device.
In Figure 5, x-axis and y-axis font size is very small. Not readable.
Author Response
Dear Reviewer
We thank You very much for Your remarks and suggestions. All Your remarks and suggestions were taken into account:
- the Introduction part with literature review was prepared
- the number of pages was increased from 7 to 13
- the number of figures: from 6 to 14
- number of references: from 14 to 47
The Reviewer’s comments |
Author’s responses |
Authors should explain further information about stacked nanowire-based field effect transistor. How it works and some review of literature. |
We agree. It was done in the Introduction. |
In line 17, the authors state that stacked NWFET construction causes the degradation of device parameters: carrier mobility μe f f , threshold voltage Vth, drain currents, Ion/Io f f. It will be better if the authors add one paragraph regarding the theoretical discussion that how these device parameters are degraded for stacked NWFT construction. |
It was done partially in the Introduction |
Authors should explain the models used in this simulation such as Charge Carrier Transport and Temperature - Hydrodynamic Model. Quantization - Density Gradient Model. Mobility Model - Enormal (Ialmob Coulomb2D) PhuMob. Ballistic Mobility - BalMob. Recombination Model - SRH (DopingDep). BandGap Narrowing Model - Slotboom. |
The additional information for the mobility models was presented. For other physical effect models the additional references were presented. |
It will be better to draw a flow chart for the simulation. |
We used the traditional TCAD methodology of simulation, so the flow chart will be not original. |
Table 1: why threshold (as well Ioff) varies so much from one device to the other device. In Figure 5, x-axis and y-axis font size is very small. Not readable. |
The Fig. 5 is improved. The Ioff values in Table 1 are the theoretical results received by simulation using physical equations build into TCAD program. For real devices the difference may be not so big but we don’t find the publications with experimental Ioff values for sub-14nm NWFET
|
With best regards, sincere Yours
Konstantin O. Petrosyants.

Reviewer 3 Report
In this work, the authors investigate the performance parameters of FinFETs and NWFETs using TCAD-simulation. The characteristics of transistors were well analyzed with experimental results and theoretical interpretations. However, some major issues should be resolved for the publication to the journal of Micromachines.
1. In the introduction part, the historical review and in-depth background for the justification of this work are very lacking. The authors need to strengthen the significance of this study through more in-depth literature review.
2. Formulas for effective mobility in experimental variables and interpretation of results should be also investigated.
3. For the optimization of VLSI characteristics, it is necessary to show the thickness conditions of the fabricated device at more various values.
Author Response
Dear Reviewer
We thank You very much for Your remarks and suggestions. All Your remarks and suggestions were taken into account:
- the Introduction part with literature review was prepared
- the number of pages was increased from 7 to 13
- the number of figures: from 6 to 14
- number of references: from 14 to 47
The Reviewer’s comments |
Author’s responses |
In the introduction part, the historical review and in-depth background for the justification of this work are very lacking. The authors need to strengthen the significance of this study through more in-depth literature review. |
The authors are agreed with this remark. We made the literature review and strengths the significance of our study in the Introduction |
Formulas for effective mobility in experimental variables and interpretation of results should be also investigated. |
Additional comments for effective mobility were presented in the page 5. The additional experimental information for ueff of NWFET is presented in Fig 6. |
For the optimization of VLSI characteristics, it is necessary to show the thickness conditions of the fabricated device at more various values. |
We agree but it is difficult to prepare the special analysis at the short time (10 days). The influence of thickness conditions on the characteristics of fabricated devices is commented in new version of the Introduction and is illustrated in Fig. 5, Fig. 11, Table 3 and corresponding References. |
With best regards, sincere Yours
Konstantin O. Petrosyants.

Round 2
Reviewer 3 Report
All concerns have been resolved in the revised manuscript.